# Facile Transformation from Rofecoxib to a New Near-Infrared Lipid Droplet Fluorescent Probe and Its Investigations on AIE Property, Solvatochromism and Mechanochromism

**DOI:** 10.3390/molecules28041814

**Published:** 2023-02-15

**Authors:** Yongbo Wei, Wei Liu, Zexin Wang, Nannan Chen, Jingming Zhou, Tong Wu, Yuqiu Ye, Yanbing Ke, Hong Jiang, Xin Zhai, Lijun Xie

**Affiliations:** 1Fujian Provincial Key Laboratory of Screening for Novel Microbial Proucts, Fujian Institute of Microbiology, Fuzhou 350007, China; 2Key Laboratory of Structure-Based Drug Design and Discovery, Ministry of Education, School of Pharmaceutical Engineering, Shenyang Pharmaceutical University, Shenyang 110016, China

**Keywords:** lipid droplets (LDs), aggregation-induced emission (AIE), near-infrared (NIR), solvatochromism, rofecoxib

## Abstract

Lipid-related cancers cause a large number of deaths worldwide. Therefore, development of highly efficient Lipid droplets (LDs) fluorescent imaging probes will be beneficial to our understanding of lipid-related cancers by allowing us to track the metabolic process of LDs. In this work, a LDs-specific NIR (*λ_max_* = 698 nm) probe, namely BY1, was rationally designed and synthesized via a one-step reaction by integrating triphenylamine (electron–donor group) unit into the structure of rofecoxib. This integration strategy enabled the target BY1 to form a strong Donor–Acceptor (D-A) system and endowed BY1 with obvious aggregation-induced emission (AIE) effect. Meanwhile, BY1 also showed observable solvent effect and reversible mechanochromatic luminescent property, which could be interpreted clearly via density functional theory (DFT) calculations, differential scanning calorimetry (DSC), powder X-ray diffraction (XPRD), and single crystal X-ray data analysis. More importantly, BY1 exhibited highly specific fluorescent imaging ability (Pearson’s correlation = 0.97) towards lipid droplets in living HeLa cells with low cytotoxicity. These results demonstrated that BY1 is a new promising fluorescent probe for lipid droplets imaging, and it might be beneficial to facilitate biological research of lipid-related cancers.

## 1. Introduction

The development of new aggregation-induced emission (AIE) luminogens (AIEgens) have attracted growing interest and increasing attention due to their promising applications in the academic and industrial fields [1,2,3,4,5,6] since the AIE concept was first proposed by Tang and his coworkers [7,8]. As the full name of AIE implies, AIEgens are accompanied by intense fluorescence in the aggregated or solid states, in sharp contrast with conventional aggregation caused quenching (ACQ) fluorophores [9,10]. Thus, due to this unique fluorescence property, a large number of AIEgens were synthesized and widely applied in various areas, such as organic light-emitting devices (OLED) [11,12], chemical sensors [13,14,15], fluorescence bioimaging [16,17], diagnosis [15,18,19], etc. [20]. Of these, fluorescent imaging of lipid droplets (LDs) has proven to be of considerable importance due to its various biofunctions, such as regulations of the storage and metabolism of neutral lipids, protein degradation, construction and maintenance of membrane, and signal transduction. Therefore, the localization and tracking of LDs are of vital importance in biomedical research and clinical diagnosis [21,22]. 

Currently, Nile Red and Bodipy 503, as two well-known commercially available LDs imaging probes, possess the disadvantages of poor specificity or small stokes shifts [23,24], respectively. In addition, these commercial LDs probes suffer from poor or totally quenched luminescence properties in aggregated states when stored in LDs, which can only be used under dilute concentrations, leading to a serious photobleaching issue [25]. Conversely, new LDs probes developed from AIEgens exhibit great potential in real-time localization and dynamic monitoring in biomedical applications, since they could be well stored in LDs and maintain bright fluorescent intensity [26]. Thereafter, several AIEgens probes for LDs monitoring were developed recently [27,28], and most of them emit short wavelengths between the blue and yellow regions. Moreover, only a few Near-infrared (NIR) AIEgens were developed; they present significant advantages in the field of LDs imaging due to their strong depth penetration and reduced tissue damage, but complicated synthetic steps [29,30]. Therefore, there is still a high demand for the development of NIR AIEgens LDs probes which could be obtained by conducting simple synthetic steps.

Herein, we synthesized a new NIR LDs probe BY1 by integrating a triphenylamine (TPA) [31,32] unit into the skeleton of rofecoxib via one facile step reaction. Specifically, the existed methyl sulfonyl group of rofecoxib acts as an electron–acceptor group, which is linked to TPA through the middle furanone lactone ring as an electron–donor group. This integration strategy not only extended the conjugated system, but also enabled the target BY1 to form a donor–acceptor (D-A) system. Next, single crystal X-ray diffraction analysis showed that there is no obvious *π*…*π* interaction in the crystal packing modes, which is mainly responsible for its obvious AIE property. In addition, we also further explored the solvent effect and reversible mechanochromatic luminescent (MCL) property. This strategy showed a powerful ability to image the LDs specifically (Pearson’s correlation: *R* = 0.97). Taken together, this work not only affords a new NIR AIEgens scaffold with a larger Stokes shift of 234 nm, but also provides a new promising LDs imaging probe in biomedical research.

## 2. Results and Discussion

### 2.1. Photophysical Property

The UV/vis absorption and emission properties of BY1 were measured in the dimethyl sulfoxide (DMSO) solution (50 μM). As displayed in Figure 1A, BY1 has a maximum absorption wavelength (*λ_abs_*) of 464 nm and a maximum emission wavelength (*λ_em_*) of 698 nm in the NIR region, with a large Stokes shift of 234 nm. The maximum absorption wavelength may be induced by the intramolecular charge transfer (ICT) [33]. Due to the formation of the D-A system, an extraordinary charge transfer from TPA branch to the methyl sulfonyl group might be easily promoted upon excitation. This implies that less energy is required during the energy-level transition, resulting in longer absorption wavelengths. 

To gain a deep insight into the optical behaviors of the BY1, density functional theory (DFT) calculation was carried out at the B3LYP/6-311++G (d, 2p) level in the Gaussian 09 suite of programs [34]. The highest occupied molecular orbital (HOMO) and the lowest unoccupied molecular orbital (LUMO) were investigated by Multiwfn [35] and VMD software [36]. As displayed in Figure 1B, the HOMO electron density is mainly distributed in the diphenyl amino region, while in LUMO, owing to the strong electron-withdrawing effect of the methyl sulfonyl group, the electron density is pulled to the lactone ring and adjacent benzene ring region. The energy gap between HOMO and LUMO was estimated to be 2.69 eV, which suggested that BY1 is easily excited due to the formation of the D-A system. These results demonstrated that the ICT and extended π conjugation system play vital roles in the NIR emission of BY1.

### 2.2. Solvatochromism

Due to its significant ICT effect, we further studied the optical properties of BY1 in different solvents. The *λ_abs_* fluctuated only slightly in response to different solvent polarities (Appendix A). As shown in Figure 1C, the color of the emission was orange in the solution of CHCl_3_ and THF, whereas in highly polar media such as EtOH, DMSO, and DMF, the intensity of emission decreased significantly. Apparently, the light emission was red-shifted and weakened in intensity by increasing the solvent polarity from DCM to DMSO, indicating that the photoluminescence intensity (PL intensity) and the *λ_em_* was highly dependent on the polarity of solvents. This phenomenon may be ascribed to the twisted intramolecular charge transfer (TICT) effect [37]. In addition, the effect of solvent polarity on the Stokes shift was described by the Lippert–Mataga equation [38], and its plot of Stokes shift against the orientation polarizability of the solvent gave an upward straight line with a moderate slope, indicative of a moderate ICT feature (Figure 1D) [39]. 

### 2.3. Aggregation-Induced Emission

Next, in order to investigate the fluorescent behavior of aggregates, we chose DMSO and water as the mixed solvent. As shown in Figure 1A and Figure 2A, when BY1 was dissolved in DMSO, a very faint red emission was observed with *λ_max_* of 698 nm. However, the emission intensity was nearly quenched with water volume fractions from (*f*_w_) 0 vol% to 40 vol% (Figure 2B) because of TICT transition [40]. The fluorescent intensity was significantly enhanced with the increase in *f*_w_ from 40 vol% to 60 vol% (Figure 2B), which was ascribed to the aggregates of the BY1. Additionally, the QY of BY1 in the *f*_w_ = 70 vol% (QY = 8.64%) demonstrated an approximately 6.13-fold increase as comparison with *f*_w_ = 0 vol% (QY = 1.41%) (Table 1). These results demonstrated that BY1 is a typical AIEgen. We also investigated the absorption (Appendix A) and emission (Figure 2C) properties in an ethanol/glycerol mixture. When the ration of glycerol volume fractions (*f*_g_) in the ethanol increased from 90% to 100%, the PL intensity showed an obvious increase. This suggested that in pure glycerol, the intramolecular rotation was restricted by the steep increase in viscosity and the non-radiative pathway was significantly reduced [41], resulting in the increase in emission intensity.

### 2.4. Single Crystal X-ray Analysis

Single crystal of compound BY1 was obtained (CCDC 2223832) in acetone via natural vaporization and its structure was determined via X-ray single crystallography. The crystallographic data were summarized in Appendix A and the Oak Ridge thermal ellipsoid plot (ORTEP) diagram in the crystal was shown in Figure 3A [42]. As a kind of monoclinic space group P2/c, the dihedral angles of BY1 between the adjacent A–D, B–D, and C–D rings were 105.90°, 165.52° and 27.97°, respectively, indicating the existence of this highly distorted conformation, which was a main factor contributing to its AIE effect [43,44]. Meanwhile, as shown in Figure 3B, the centroid distance of benzene rings between adjacent molecules reached 4.115 Å without evident π…π interactions. In addition, C-H…O (2.600 Å, 2.698 Å), and C-H…π (2.859 Å, 3.252 Å, 3.372 Å) interactions could be observed (Appendix A) between the adjacent molecules, which might be also beneficial for its bright emission in solid. 

Furthermore, Hirshfeld surface analysis [45] was also carried out based on the single crystal X-ray data (Figure 3C). The red spots appeared on the d_norm_ surface of BY1 revealed the existence of C-H…O and C-H…π, in agreement with intermolecular interactions in Figure 3D. In addition, as displayed in Figure 4, different types of intermolecular interactions were presented in two-dimensional fingerprints, and their specific contribution percentages were as follows: H…H (57.6%) > H…C/C…H (21.5%) > H…O/O…H (16.1%). Notably, the C…C inter-atomic contacts comprise only 3.4% of the whole crystal packing modes, indicating that the C…C interaction does not play a major role in the crystal stacking of compound BY1. This is consistent with the previous result in Figure 3B. 

### 2.5. Mechanochromic Properties

Based on the AIE activity of BY1, the MCL property was also investigated by grinding the powders of BY1. As shown in Figure 5 and Table 2, the pristine powders showed bright red-orange emission (*λ_max_* = 626 nm). Once grinding was performed in a mortar and pestle, the emission red-shifted to the red (*λ_max_* = 658 nm), indicating MCL behaviors. Meanwhile, to check the MCL reversibility of BY1, the ground powders were heated at 70 °C within 30 min. The ground powders could be transferred into red-orange emission, which is similar to that of the pristine solids (Figure 5A). Meanwhile, the corresponding maximum emission peak returned to the position of the initial pristine wavelength. Interestingly, when immersed in acetone for 1 min, the maximum emission wavelength was blue-shifted compared to the pristine powders, indicating a reversibility from ground powders to immersing powders (Figure 5B). Moreover, grinding–immersing and grinding–heating processes could be performed reversibly several times without obvious fatigue (Figure 5C–F), which suggested that compound BY1 possessed good mechanochromic properties.

To obtain a deep understanding of the mechanism of MCL phenomena, PXRD was investigated (Figure 5G,H). The pristine of BY1 displayed a crystalline state with sharp peaks. After grinding, the diffraction peaks disappeared, which indicated a loss of crystallinity for the ground sample. When we heated or immersed the ground sample, some small diffraction peaks reappeared, which demonstrated that grinding samples could be restore them to their original crystallinity. To confirm the transition from a crystalline structure to an amorphous state upon grinding, DSC experiments were also performed. In a DSC measurement of grinding BY1, exothermic peaks that corresponded to the cold-crystallization transition (T = 181.6 °C) were observed, followed by endothermic peaks that corresponded to the melting point of powder BY1 (T = 229.3 °C) [46]. These experimental results suggested that the transformation between crystalline structure and the amorphous state was responsible for the observed MCL behavior. 

### 2.6. Lipid Droplet Imaging

To explore the application of BY1 in LDs imaging in living cells, the potential cytotoxicity was evaluated using the Cell Counting Kit-8 (CCK8) assay and low cytotoxicity was observed (Appendix A). Next, the LDs imaging in HeLa cells was investigated via CLSM to confirm the specificity of the BY1 for LDs and the co-localization experiment was conducted by incubating HeLa cells with BY1 and Bodipy 503 (Figure 6). As illustrated in Figure 6E, BY1 selectively accumulated in LDs and displayed bright red fluorescence. Additionally, the staining pattern of BY1 was almost identical to the staining results of Bodipy 503 (Figure 6B–D). Moreover, a high overlap of the intensity profiles of BY1 and Bodipy 503 in the region of interest (ROI) was observed, and the intensity changes were closely synchronized (Figure 6G), which suggests a high overlap between BY1 and Bodipy 503. The Pearson’s correlation of the green channel (Figure 6B) and the red channel (Figure 6E) was calculated with a value up to 0.97 [22], indicating the powerful LDs-specific targeting capability. These results support the theory that BY1 is a promising NIR fluorescent probe for LDs imaging.

## 3. Materials and Methods

### 3.1. General Information and Materials

All commercial reagents and solvents were used as received. Rofecoxib was purchased from Changzhou Yabang Pharmaceutical Co., LTD. (China). 4-(*N*,*N*-diphenylamino)benzaldehyde was purchased from Accela ChemBio Co., Ltd. (Shanghai, China). Reactions were magnetically stirred and monitored on a HP-TLC Silica Gel 60 GF254 plate from Shanghai Haohong Biomedical Technology., LTD (Shanghai, China). All other reagents and solvents were purchased from Sigma-Aldrich (Burlington, MA, USA) and used without further purification, unless otherwise stated. Fluorescence spectra were recorded on a Varioskan LUX 3020-80110. Differential scanning calorimetry (DSC) traces were collected on a DSC STAR system at a heating rate of 10 °C/min from 40 °C to 300 °C under a high-purity nitrogen atmosphere. Powder X-ray diffraction (PXRD) patterns were recorded on a Rigaku (D/MaX-3B) diffractometer. Single-crystal X-ray diffraction was conducted with CrysAlisPro 1.171.40.39a (Rigaku OD, 2019) with *λ* = 0.77 Å (MoKα). The fluorescence quantum yield (QY) was measured on Edinburgh FLS980 steady-state transient fluorescence spectrometer equipped with a built-in integrating sphere. (Parameters: The light source was Xenon Xe1, the scan step was 1 nm, the dwell time was 0.2 s, and a total of 3 scans were carried out.) LDs imaging was performed with a Nikon Ti-E&C2 scanning unit. ^1^H NMR and ^13^C NMR spectra were measured on a Bruker AV 600 spectrometer in appropriated deuterated dimethyl sulfoxide solution at room temperature with the solvent residual proton signal as a standard. High-resolution mass spectra (HRMS) were recorded on a GCT premier CAB048 mass spectrometer operating in MALDI-TOF mode.

### 3.2. Synthesis

In a 100 mL round-bottom flask, rofecoxib (0.200 g, 0.636 mmol) and 4-(*N*,*N*-diphenylamino) benzaldehyde (0.200 g, 0.732 mmol) were dissolved in 10 mL methanol. Then, two drops of piperidine were added and the mixture was stirred at room temperature for 12 h in a dark atmosphere. After completion of the reaction, the mixture was filtered and washed with methanol on a filter to afford the BY1 with high purity. The synthetic route is shown in Figure 1. The structure was characterized and confirmed by ^1^H NMR, ^13^C NMR, HRMS (ESI): *m*/*z* [M]^+^ calcd for C_36_H_27_NO_4_S^+^: 569.1661; found: 569.1654. (Appendix A) and X-ray structural analysis.

(*Z*)-5-(4-(diphenylamino)benzylidene)-4-(4-(methylsulfonyl)phenyl)-3-phenylfuran-2(5*H*)-one. ^1^H NMR (600 MHz, DMSO-*d_6_*) δ 8.07–8.03 (m, 2H), 7.72–7.66 (m, 4H), 7.38–7.31 (m, 9H), 7.15 (tt, *J* = 7.4, 1.1 Hz, 2H), 7.12–7.08 (m, 4H), 6.94–6.90 (m, 2H), 6.00 (s, 1H), 3.30 (s, 3H). ^13^C NMR (151 MHz, DMSO-*d_6_*) δ 168.2, 149.4, 148.9, 146.7, 146.4, 142.2, 135.7, 132.5, 130.9, 130.4, 129.6, 129.5, 129.0, 128.1, 126.5, 125.9, 124.7, 121.3, 113.7, 43.8.

### 3.3. Fluorescent Microscope Analysis

Procedures of cell culture and live cell staining: Hela cells were seeded in RPMI-1640 medium at the density of 1 × 10^5^ cells per dish with Dulbecco’s Modified Eagle Medium (DMEM) containing 10% fetal bovine serum (FBS) and a mixture of 0.1 mg/mL streptomycin and 100 units/mL penicillin, and then allowed to attach overnight at 37 °C under 5% CO_2_. Hela cells were incubated with oleic acid (0.1 mM) for 24 h, and then stained with BY1 and BODIPY503 (10 µM) for 30 min. Fluorescence images were visualized by Nikon Ti-E&C2 scanning unit.

## 4. Conclusions

In summary, the NIR fluorescent probe BY1 was rationally designed and synthesized by integrating a TPA unit into rofecoxib via a one-step reaction. BY1 exhibited significant AIE property with a large Stokes shift of 234 nm. In addition, the mechanism of AIE was elucidated clearly by analyzing the single crystal X-ray data and Hirshfeld surfaces calculation. Moreover, the mechanochromic behavior could also be interpreted by these analyses of powder PXRD and DSC. Finally, BY1 exhibited highly specific (Pearson’s correlation: *R* = 0.97) imaging ability toward LDs in living HeLa cells. This work would be beneficial for guiding the future design of new NIR AIEgens. It also affords a novel NIR probe with high specific imaging ability to LDs.

## Data Availability

Data will be subjected to availability on request to the corresponding author.

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
