# Peer review of "Facile Transformation from Rofecoxib to a New Near-Infrared Lipid Droplet Fluorescent Probe and Its Investigations on AIE Property, Solvatochromism and Mechanochromism"

_molecules, 2023, doi:10.3390/molecules28041814_

Round 1

Reviewer 1 Report

In this manuscript, By integrating triphenylamine (electron-donor group) unit into the structure of rofecoxib, the author reasonably designed and synthesized LDs-specific NIR probe BY1 through a one-step reaction, and clearly explained the AIE properties and mechanochromic luminescence properties of BY1 through density functional theory (DFT) calculations, differential scanning calorimetry (DSC), powder X-ray diffraction (XPRD) and single crystal X-ray data analysis. Compared with most LDs imaging probes reported and developed, BY1 has longer near-infrared (NIR) emission, AIE properties and MCL properties, and can explain the AIE mechanism through test data analysis. This paper reports that a new promising NIR LDS probe by1 for lipid droplet imaging was obtained through a simple synthetic step and may help facilitate the biological research of lipid related cancers. Therefore, I suggest it be accepted after the author complements the content, articulates the obscure content, and corrects the format.

1.      The introduction writing logic is less smooth. Fox example, introduction does not elaborate what is the conventional lipid droplet approach and what are its advantages and disadvantages? What are the advantages of the methods provided by the article over conventional methods?

2.      Article innovation points are not clear enough, and it is recommended to highlight the innovativeness of the article in the end section of the introduction.

3.      It is recommended that authors insert a synthetic scheme diagram in the Section 2.2. Synthesis.

4.      Figure 5. (A) (B) material color changes are not obvious, and it is recommended that the author reframe from photographing or make color labeling.

5.      Figure 6. Lipid droplet imaging interpretation is not sufficiently clear, and it is recommended that the authors contributed to figure 6 in the section 3.6. Lipid droplet imaging A more detailed explanation is made in the lipid droplet imaging section.

6.      Stokes shift units in lines 109 and 246 are suggested to be exchanged into nm, which is more helpful for the reader to understand.

7.      Reference formatting is not uniform, e.g. Articles 5, 20, 28 of the references are incorrectly formatted.

8.      Figure 2. In the caption, (b) figure explaining sentence tail missing full stops.

9.      Details on the methods used for evaluating the PLQY and the cell imaging should be provided in the experimental section;

10.  Absorption spectra in solution should be discussed and analysed in more detail

Reviewer 2 Report

Wei et al. have developed a fluorescent probe for staining LDs based on a donor-acceptor system. The probe enabled the detection of LDs in the NIR region. Moreover, this molecule shows aggregation-induced emission and reversible mechanochromatic luminescent properties as well as solvatochromism. Although each result is well discussed, it is not clear how the AIEgen properties related to the staining ability of the LDs. Does the probe aggregate in LDs? However, based on the solvatochromic features shown in Figure 1, this compound may not be classified as an AIEgen molecule because it exhibited strong emission even when dissolved in solutions, such as DCM, chloroform and THF. Although current version is not suitable for publication, I would like to recommend that this manuscript should be accepted after major revision. Before resubmission, please clarify the definition of the “AIEgen molecule” and mention how such properties are utilized for staining LDs.

Round 2

Reviewer 2 Report

I am satisfied with the revisions the authors have made. This manuscript can be accepted at present form.